# Mechanism of Magnetic Flux Leakage Detection Method Based on the Slotted Ferromagnetic Lift-Off Layer

**DOI:** 10.3390/s22093587

**Published:** 2022-05-09

**Authors:** Jian Tang, Rongbiao Wang, Gongzhe Qiu, Yu Hu, Yihua Kang

**Affiliations:** 1School of Mechanical Science and Engineering, Huazhong University of Science and Technology, Wuhan 430074, China; tangj@hust.edu.cn (J.T.); qiugongzhe@hust.edu.cn (G.Q.); huyuhy@hust.edu.cn (Y.H.); 2Key Laboratory of Nondestructive Testing, Ministry of Education, Nanchang Hangkong University, Nanchang 330063, China; wangrongbiao@nchu.edu.cn

**Keywords:** magnetic flux leakage (MFL) testing, permeability perturbation, slotted ferromagnetic lift-off layer (SFLL), leakage magnetic field (LMF) enhancement, lift-off tolerance

## Abstract

Magnetic flux leakage (MFL) testing is widely used in non-destructive testing of ferromagnetic components. In view of the serious attenuation of the leakage magnetic field (LMF) caused by the transmission of LMF in the lift-off layer between the measuring point and the workpiece, this paper introduces an MFL detection method based on the slotted ferromagnetic lift-off layer (SFLL). The conventional non-ferromagnetic lift-off layer is changed to a ferromagnetic lift-off layer with a rectangular slot. The magnetic sensor is fixed above the slot and scans the workpiece together with the lift-off layer. First, the detection mechanism of the new method was studied by an equivalent LMF coil model. The permeability perturbation effect and the magnetization enhancement effect were analyzed in the new method. Based on the detection mechanism, the lift-off tolerance of the new method was investigated. Then, the LMF enhancement and lift-off tolerance of the new method in the steel plate detection model were studied. Finally, experiments were conducted to compare the new method with the conventional method. The simulation and experimental results show that the slotted ferromagnetic lift-off layer enhances the amplitude of the MFL signal and is tolerant to the lift-off value. This method provides a new idea for optimizing the design of the MFL sensor and improving the sensitivity of MFL detection at a large lift-off value.

## 1. Introduction

Ferromagnetic components, such as steel pipes and steel plates, are key industrial components, which are widely applied in petrochemical, high-pressure equipment, thermal equipment, and engineering structures. The international standard stipulates those important ferromagnetic components must undergo non-destructive testing (NDT) after manufacture. Common NDT methods include eddy current testing (ECT) [1,2], ultrasonic testing (UT) [3,4], magnetic particle inspection (MPI) [5,6], and magnetic flux leakage (MFL) testing [7,8,9,10]. ECT has the advantages of high sensitivity and non-contact. However, due to the skin effect, it can only detect surface defects and has strict requirements on surface quality. UT can detect internal defects but requires couplant. MPI has high sensitivity and can image defects directly, but the level of automation is not high. MFL has the advantages of low cost, no pollution, fast speed, and a high level of automation. In addition, internal and external defects can be detected by MFL. Therefore, it is widely used in automated NDT of ferromagnetic components [11,12,13].

In MFL detection, the lift-off value is one of the most important parameters. The amplitude of the leakage magnetic field (LMF) decreases rapidly with the increase of the lift-off value, which leads to a decrease in detection sensitivity [14,15]. Therefore, how to improve the detection sensitivity at a large lift-off has been a key issue in MFL detection. Many scholars have also conducted related research.

To improve the detection sensitivity at a large lift-off, many highly sensitive magnetic sensitive elements have been applied in MFL detection, such as high-sensitivity Hall element, anisotropic magnetoresistance (AMR) [16], giant magnetoresistance (GMR) [17], and tunnel magnetoresistance (TMR) [18]. Meanwhile, the new sensor design also improves the sensitivity of the MFL detection. Sun Y. added an external magnetic source above the probe. Through the mutual interference between the magnetic field generated by the external magnetic source and the LMF of the defect, the sensitivity of the sensor at large lift-off was improved [19]. Wang R. added an AC excitation coil next to the induction coil sensor. The high-frequency magnetic field generated by the AC coil was used to modulate the low-frequency LMF of the defect. This method enhances the detection sensitivity at large lift-off [20]. Lee J. arranged the sensor array on the surface of the semicircular ferrofluid and concentrated the LMF through the semicircular ferrofluid with high permeability, which improves the sensitivity of the probe [21]. Wu J. proposed a lift-off tolerant probe based on the “magnetic field focusing effect”. A ferrite core was embedded in an induction coil to increase the leakage magnetic flux in the induction coil [22]. The existing literature on the MFL detection sensitivity at a large lift-off is extensive and focuses particularly on the sensor. The lift-off layer is the area between the sensor and the workpiece. No matter what method, the LMF must pass through the lift-off layer when it is transmitted to the measuring point, but the lift-off layer is always ignored in the previous research. In the conventional MFL method, the lift-off layer is always air or other non-ferromagnetic materials, as shown in Figure 1. The LMF decays rapidly in the high magnetoresistance non-ferromagnetic lift-off layer. It may result in failure to detect tiny cracks.

Some scholars believe that placing an iron plate on the workpiece will cause a “magnetic shielding effect” [23,24] for the LMF, so the ferromagnetic lift-off layer is rarely applied in MFL testing. However, Wu J. proposed an MFL testing method based on the “magnetic guiding effect”. Experiments show that under low magnetization, the ferromagnetic lift-off layer will weaken the LMF at the measuring point. However, under strong magnetization, the LMF at the measuring point increased by about 38% [25]. Previously, we have introduced an MFL method based on the ferromagnetic lift-off layer with a through groove [26]. In this paper, we further analyze the effect of the slotted ferromagnetic lift-off layer (SFLL) on the LMF. Furthermore, we focus on exploring the mechanism of the SFLL-MFL method and its lift-off tolerance. Compared with the conventional air/non-ferromagnetic lift-off layer, it can obtain a stronger MFL signal and is more tolerant to the change of the lift-off layer thickness by using the SFLL-MFL method. This method provides a new idea for improving the sensitivity in large lift-off MFL detection and has the potential to be used in combination with previous other methods to obtain a greater MFL signal further.

## 2. Mechanism

### 2.1. MFL Detection Method Based on the Slotted Ferromagnetic Lift-Off Layer

Although the ferromagnetic lift-off layer has a certain enhancement effect on the LMF at a strong magnetization, the “magnetic shielding effect” cannot be eliminated. In order to eliminate the adverse effects of the “magnetic shielding effect” and enhance the strength of the LMF further, the SFLL can force the LMF to leak out through the lift-off layer to the measuring point, as shown in Figure 2.

Based on the ferromagnetic lift-off layer, a rectangular slot is manufactured. The sensor is fixed at the measuring point on the slot and scans the workpiece with the lift-off layer, as shown in Figure 2a. Different from the conventional air lift-off, the measured magnetic field at the measuring point in the new method includes the LMF of the defect and the slot. The measured magnetic field at the measuring point is defined as the measuring point magnetic field. In the new method, the actual defect LMF is the difference between the measuring point magnetic field with and without defects.

Under the magnetizing field, the slot will generate an LMF. When a defect is detected, as shown in Figure 2b, the LMF of the slot leaks upward into the air where the sensor is and also spreads downward into the workpiece where the defect is. The slot LMF diffused down increases the magnetization of the defect. Meanwhile, for the lift-off layer, the LMF generated by the defect also enhances the magnetization of the slot compared to the absence of defects. The enhanced magnetization of the defect and the slot means that they also generate stronger LMF. Therefore, compared with the non-ferromagnetic lift-off layer, the SFLL can enhance the LMF of defects.

### 2.2. The Mechanism of LMF Enhancement

According to the above, the new method is based on the enhancement of magnetization to increase the LMF, so magnetization is a key factor to study the enhancement mechanism. However, variables cannot be controlled singly due to the coupling between the defect LMF and the magnetization of the SFLL. Because of the similarity of the spatial distribution of the LMF and the magnetic field near the coil, the magnetic field generated by an equivalent LMF coil in the horizontal direction was used to replace the defect LMF to control a single variable. When the equivalent LMF coil is energized, it is equivalent for a defect; if not, it is equivalent for no defects.

Stationary analysis was carried out with a two-dimensional finite element model. This model is composed of a slotted iron plate, a magnetizing coil, and an equivalent LMF coil. The magnetizing coil produces a horizontal magnetization field in the slotted iron plate. The equivalent LMF coil produces a local magnetization field around the slot. The two magnetization fields have the same direction in the slotted iron plate. The relevant dimensions are shown in Figure 3. The width of the slot was set as 0.5 mm, and the measuring point was 0.3 mm away from the upper surface of the slotted iron plate (SFLL). The equivalent coil for the defect had 100 turns and a current of 0.5 A. The magnetization coil had 1000 turns, whose current was increased from 0.5 A to 15 A with an interval of 0.5 A. At the same time, the thickness of the iron plate *t* was also increased from 0.5 mm to 3 mm with an interval of 0.5 mm. The slotted iron plate (Q235) was considered to be isotropic ferromagnetic material. The *B*-*H* curve of carbon steel Q235 is shown in Table 1. The mesh type was free triangular. In order to get better results, the fine mesh was generated within the slot and measuring point region. The boundary conditions between air and ferromagnetic materials meet the continuity conditions of magnetic field and magnetic induction intensity. The outer boundary of the air domain satisfies the boundary condition that the magnetic flux is parallel to the boundary.

Simulation models corresponding to each parameter were simulated separately to obtain the magnetic induction intensity *B*_on_ at the measuring point. Then the magnetic induction intensity *B*_off_ is subtracted at the measuring point when the equivalent LMF coil is not energized under the corresponding model. The defect LMF under different SFLL thicknesses is obtained. The simulation results are shown in Figure 4.

The simulation results show that when *t* < 1.5 mm, the LMF is almost unchanged at first, then increases rapidly with the increase of magnetization, and finally it gradually decreases after reaching the maximum value. When *t* > 1.5 mm, the LMF is almost constant at first and then decreases slowly with the increase of the magnetization. It can be seen that the magnetization effect of the equivalent LMF coil on the SFLL is inconsistent under different plate thicknesses. Therefore, further analysis is needed.

The defect LMF *B*_MFL_ is the difference between the measuring point magnetic field when the equivalent LMF coil is energized *B*_on_ and powered off *B*_off_. The magnetic field of the equivalent LMF coil enhances the local magnetization of the slotted iron plate. The magnetization near the slot is enhanced by a certain Δ*H*, and the corresponding LMF of the slot is enhanced by Δ*B*. Then the change of the LMF of the slot under different magnetizations was simulated as shown by the black line in Figure 5. The increasing value of the LMF Δ*B* gradually decreases with the increase of the magnetizing current. This characteristic quantity can be represented by the derivative of the LMF intensity with respect to the magnetizing current $B_{\mathrm{MFL}}^{\prime }$ as shown by the red line in Figure 5. The derivative of the LMF intensity $B_{\mathrm{MFL}}^{\prime }$ decays rapidly with the increase of the magnetizing current at first, and then slowly decays to 0, which explains the decay trend in the latter part of the curves in Figure 4. The increase of the LMF in the SFLL-MFL method due to the magnetization enhancement of the slot is called the magnetization enhancement effect in this paper.

The magnetic field generated by the equivalent LMF coil produces magnetization enhancement in the local area of the iron plate. The relative permeability of the ferromagnetic material will also change according to the *B-H* curve under different magnetizations. In other words, permeability perturbations are generated in the localized regions of enhanced magnetization [27]. Under the magnetization of the external magnetic field, the permeability perturbation area will also cause magnetic refraction [28], thereby generating an additional LMF. The enhancement of the LMF at the measuring point due to the permeability perturbation is called the permeability perturbation effect in this paper.

To verify the permeability perturbation effect, the length of the equivalent LMF coil was gradually increased based on the simulation model shown in Figure 3. Lengthening the equivalent LMF coil will make the permeability perturbation area of the iron plate away from the vicinity of the slot. The magnetic field near the defect became gradually uniform, thereby eliminating the influence of the permeability perturbation effect. The lengths of the equivalent LMF coils were 2 mm, 4 mm, 6 mm, 8 mm, 10 mm, and 20 mm, respectively. The simulation results are shown in Figure 6. The results show that as the length of the equivalent LMF coil increases, the variation trend of the equivalent defect LMF with the magnetizing current gradually changes from increasing first and then decreasing to decreasing all the time, which is consistent with the variation trend of the derivative of LMF intensity in Figure 5.

The result of the 20 mm long coil is compared with the derivative of LMF intensity, as shown in Figure 7. It can be found that the change trend is almost the same. As a result, the upward trend in the first half of each curve in Figure 4 is caused by the permeability perturbation effect.

The above simulation results show that the enhancement of the defect LMF in the SFLL-MFL method is caused by the compounding of the magnetization enhancement effect and the permeability perturbation effect.

### 2.3. Lift-Off Tolerance Mechanism of SFLL-MFL Method

Based on the principle of the SFLL-MFL method, when the thickness of the lift-off layer increases from *t*_1_ to *t*_2_, the lift-off value of the sensor increases, as shown in Figure 8. The LMF generated by the permeability perturbation effect will decay rapidly with the increase of the thickness like the detection of internal defects in the conventional MFL method [27].

However, the distance between the sensor and the lift-off layer does not change. The signal contributed by the LMF of the slot does not decrease with the increase of the thickness of the lift-off layer. Moreover, the depth of the slot is equal to the thickness of the lift-off layer, so as the thickness of the lift-off layer increases, the depth of the slot also increases. Therefore, the effect of increasing the lift-off layer thickness on the magnetization enhancement effect needs further study.

The characteristic quantity of the magnetization enhancement effect is the derivative of the LMF of the slot with the magnetization. Therefore, keeping other conditions unchanged, the relationship between the LMF intensity and the magnetizing current for different thicknesses of the SFLL was simulated. The thickness of the SFLL was increased from 0.5 mm to 3.0 mm with an interval of 0.5 mm. The magnetizing current was increased from 0.5 A to 15 A with an interval of 0.5 A. The derivation of the slot LMF with iron plates $B_{\mathrm{MFL}}^{\prime }$ of different thicknesses was obtained separately. The simulation results are shown in Figure 9.

It can be seen that each curve drops quite slowly at first, then falls rapidly, and finally tends to be flat. The relationship between the derivative of LMF intensity and the thickness of SFLL is observed. Taking *t* = 1.0 mm and *t* = 1.5 mm as examples, the derivative of LMF begins to decrease rapidly when the magnetizing current exceeds the initial gentle decline stage in the curve of *t* = 1.5 mm, but the value of the *t* = 1.5 mm curve is still greater than that of the *t* = 1.0 mm curve under the same magnetizing current, as shown in the yellow area in Figure 9. This shows that when the magnetization exceeds the stage where the LMF increases linearly with the magnetization, as the thickness of the SFLL increases, the magnetization enhancement effect will increase accordingly.

In summary, when the thickness of the lift-off layer increases, compared with the conventional MFL method, the magnetization enhancement effect in the SFLL-MFL method compensates for the attenuation of the LMF. Therefore, the new method is tolerant to the increase of lift-off value, as shown in Figure 10.

## 3. Simulation

The magnetization enhancement effect and permeability perturbation effect in the SFLL-MFL method and the lift-off tolerance were analyzed using the equivalent LMF coil model. Next, the MFL detection model for the steel plate workpiece was used to verify the detection effect of the SFLL-MFL method. The two-dimensional simulation model shown in Figure 11 was established by using the COMSOL simulation software.

This model is composed of a steel plate workpiece, a slotted iron plate, and a magnetizing coil. The magnetizing coil produces a horizontal magnetization field in the slotted iron plate and the workpiece. There was a groove defect on the steel plate workpiece. The slotted iron plate lift-off layer was located on the steel plate workpiece. The scanning path of LMF was 0.3 mm above the lift-off layer. The relevant dimensions are shown in Figure 11 and the specific parameters are shown in Table 2. Other technical information about the simulation model is consistent with Section 2.2.

### 3.1. The Enhancement of the LMF by the SFLL

First, the detection effect of the three different kinds of lift-off layers of slotted iron plate, iron plate, and air on the LMF was verified. The LMF distributions of the defect were simulated under three conditions, respectively. Magnetic induction intensity in the horizontal direction *B_x_* was extracted along the scanning path, and the background magnetic fields of the corresponding three models without defects were subtracted to obtain the distribution of the defect LMF, which is plotted in Figure 12. It can be seen that the LMF intensity of the slotted iron plate lift-off layer was 79 mT, the iron plate lift-off layer was 30 mT, and the air was 24 mT. The simulation results verified that the iron plate lift-off layer has a certain enhancement effect relative to the air lift-off layer, but is not prominent. When compared with the air lift-off layer, the LMF amplitude *B_x_* with the slotted iron plate lift-off layer was enhanced by 263%. The enhancement effect of the SFLL is remarkable.

### 3.2. Lift-Off Tolerance of the SFLL-MFL Method

The thickness of the lift-off layer *t* varied from 0.25 mm to 3 mm with an interval of 0.25 mm. The amplitudes of the LMF *B_x_* were plotted in Figure 13. It can be seen that the *B_x_* of the LMF of the three kinds of lift-off layers are all attenuated with the increase of the lift-off layer thickness. However, compared with the air lift-off layer, the iron plate lift-off layer only enhances the LMF when *t* is less than 1.5 mm. When *t* is greater than 1.5 mm, the LMF is weakened instead of enhanced. The *B_x_* of the slotted iron plate lift-off layer is stronger than that of the iron plate lift-off layer and the air lift-off layer at different *t*.

The three curves were normalized and the attenuation speeds of the three curves were observed, as shown in Figure 14. The attenuation percentages of LMF amplitudes for three kinds of lift-off layers were extracted and the thicknesses of the lift-off layers ranged from 0.25 mm to 1 mm, 0.25 mm to 2 mm, and 0.25 mm to 3 mm, respectively, as shown in Table 3. It can be seen that when the lift-off value increases from 0.25 mm to 1 mm, the *B_x_* of both the iron plate lift-off layer and the air lift-off layer are declined by 62.1% and 61.8%, respectively, while that of the slotted iron plate is only declined by 16.9%. When the *t* increases to 2 mm, the *B_x_* of both the iron plate and the air lift-off layer have been attenuated by 89.6% and 81.6%, respectively, while attenuation amplitude with the slotted iron plate has not exceeded 30.2%. Thickened to 3mm, the SFLL-MFL method is still relatively lift-off tolerant.

## 4. Experimental Verification

In order to verify the feasibility of the SFLL-MFL method, an experimental platform was built as shown in Figure 15. The experimental sample was a Q235 steel plate with a length of 300 mm, a width of 40 mm, and a thickness of 10 mm. An artificial defect of the transverse rectangular groove was processed by electrical discharge machining on the upper surface of the steel plate sample. The groove defect length was 40 mm, the width was 0.6 mm, and the depth was 1 mm. The DC magnetizing coil with 3000 turns was powered by a DC magnetizing power source. The steel plate sample was longitudinally magnetized with a magnetizing current of 10 A. The Hall element HG-0811 was adopted as the magnetic sensor, and the signal was collected and displayed by the oscilloscope directly. To ensure the accuracy of the slotted iron plate and copper plate lift-off layer thickness, feeler gauges were used as the lift-off layer for the experiments. The thickness of the two kinds of lift-off layers was set to 1.5 mm. The slot on the iron plate was processed by wire cutting, and the slot width was 0.6 mm.

The Hall element was fixed on the slotted iron plate and aligned with the centerline of the slot. The slotted iron plate, which was close to the steel plate sample, scanned the defect on the steel plate sample several times. Stable MFL signal waveforms were obtained on the oscilloscope. Then the Hall element was fixed on the copper plate and scanned the defects in the same way. The signal waveforms of the copper plate were obtained on the oscilloscope. The experimental results are shown in Figure 16. It can be seen that the signal amplitude of the copper plate lift-off layer is 29 mV, while the signal amplitude of the slotted iron plate lift-off layer is 98 mV, which is about 3.4 times that of the copper plate lift-off layer. The enhancement effect of the MFL signal in the SFLL-MFL method is obvious.

Then, the effect of the slotted iron plate and the copper plate on the intensity of the LMF was verified under different lift-off layer thicknesses. For verification, 0.2 mm, 0.4 mm, 0.6 mm, 0.8 mm, 1.0 mm, 1.5 mm, 2.0 mm, 2.5 mm, and 3.0 mm thick copper plate lift-off layers and slot iron plate lift-off layers were used. The amplitudes of the MFL signals under the above-mentioned experimental conditions are obtained, respectively. The average value was obtained by scanning multiple times, as plotted in Figure 17.

The experimental results show that with the increase of the thickness of the lift-off layer, the amplitudes of the MFL signals of the two kinds of lift-off layers both decrease gradually, but the amplitude of the MFL signals of the slotted iron plate is higher than that of the copper plate. When the lift-off layer thickness is 0.2 mm, the signal amplitude of the slotted iron plate is 1.32 times that of the copper plate. As the thickness increases, the magnification with the slotted iron plate gradually increases to 5.6 times. The enhancement effect is remarkable.

In addition, the attenuation speed of the slotted iron plate is also slower than that of the copper plate. In the process of increasing the lift-off layer thickness from 0.2 mm to 3 mm, the signal amplitude with the copper plate is attenuated by 87%, while the signal amplitude of the slotted iron plate is only attenuated by 46%.

Compared with the simulation results shown in Figure 13, the copper plate in the experiment is equivalent to the air lift-off layer in the simulation. It should be noted that the vertical axis label of simulations is LMF amplitude *B_x_* and the vertical axis label of experiments is the signal amplitude of the Hall sensor. The values of the simulation and experiment results cannot be compared directly. Even so, we can find that the SFLL and non-ferromagnetic lift-off layer have the same attenuation trend. For the attenuation speed of the non-ferromagnetic lift-off layer, the simulation result is 88.5% and the experiment result is 87%. There is little difference between the two results. For the attenuation speed of SFLL, the simulation result is 39.3% and the experiment result is 46%. The two results are not completely consistent. This is mainly because there is a difference in the magnetic properties between the experimental steel plate sample and the model in the simulation software. In addition, the position of the sensor on the lift-off layer may have a certain error. However, the numerical difference does not affect the laws and conclusions.

Then, the magnetization current *I* varied from 6 A to 14 A with an interval of 2 A, and the thickness of the slotted iron plate and copper plate changed from 0.25 mm to 3 mm. The attenuation speed of two lift-off layers was compared in Table 4. What stands out in this table is the slow attenuation speed with the slotted iron plate compared with the iron plate or copper plate. When the magnetization current *I* increased from 6 A to 14 A, the attenuation speed with the copper plate was more than 80% and the attenuation speed of the copper plate was practically constant with the change of magnetizing current. While the attenuation speed with the slotted iron plate varied from 41.94% to 52.6%, which is much slower than that of the copper plate. Meanwhile, what can be seen is the growth of the attenuation speed with the increase of the magnetization current. According to Section 2.2, the magnetization enhancement effect decays with the increase of the magnetizing current. Therefore, with the growth of the magnetization current, the magnetization enhancement effect decays and the attenuation speed with the slotted iron plate increases. The experimental results are consistent with the previous mechanism analysis.

Therefore, it has been verified that compared with the non-ferromagnetic lift-off layer, the SFLL is more tolerant to the change of the lift-off value.

## 5. Conclusions

In this paper, an MFL detection method based on the slotted ferromagnetic lift-off layer is introduced. Compared with the conventional non-ferromagnetic lift-off layer, the SFLL-MFL method can obtain a stronger MFL signal with a large-thickness lift-off layer. In the experiment with a 3 mm thick lift-off layer, the signal amplitude of the new method is 5.6 times that of the conventional MFL method. The enhancement effect of the new method comes from the magnetization enhancement effect and the permeability perturbation effect. The magnetization enhancement effect makes the new method more tolerant to the change in the lift-off layer thickness. In the experiment of increasing the lift-off layer thickness from 0.2 mm to 3 mm, the signal amplitude of the copper plate is attenuated by 87%, while that of the slotted iron plate is only attenuated by 46%. Whether the lift-off layer is thickened or the lift-off layer is worn and thinned during working, a stronger and more stable signal can be obtained.

The implementation of the SFLL-MFL method is simple and effective, which has good engineering application potential for MFL detection of contactable ferromagnetic workpieces. In the next work, we are going to make the SFLL with flexible magnetic material or magnetic fluid, so as to adapt to the MFL detection of complex parts, which is not accessible to the probe, and precision parts that are forbidden to be contacted by the solid probe.

## Figures and Tables

**Figure 1 sensors-22-03587-f001:**
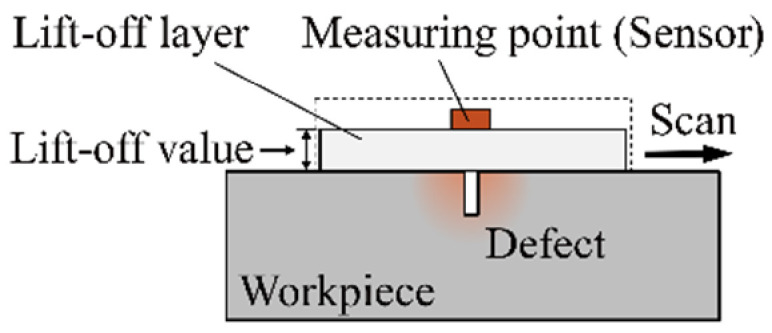
The lift-off layer and lift-off value in MFL testing.

**Figure 2 sensors-22-03587-f002:**
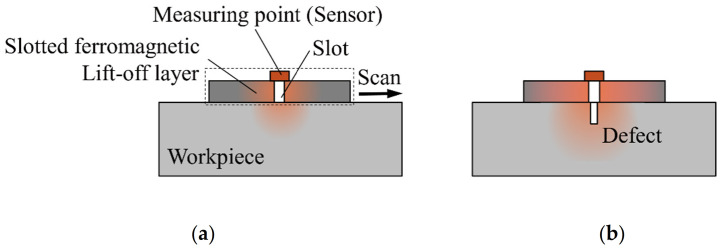
The method of the slotted ferromagnetic lift-off layer (**a**) without defects and (**b**) with a defect in the workpiece.

**Figure 3 sensors-22-03587-f003:**
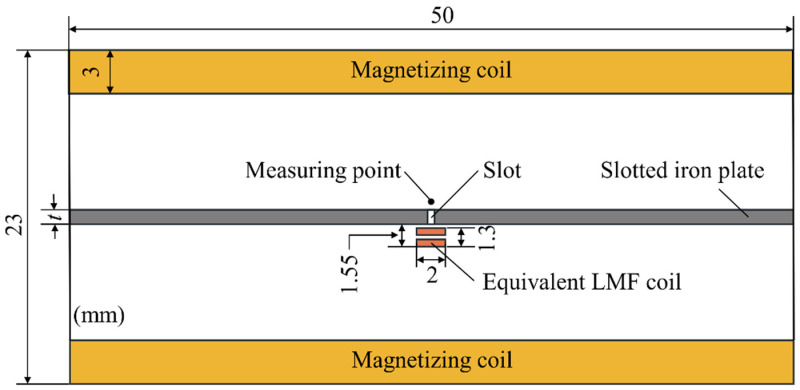
Simulation model for the mechanism of LMF enhancement.

**Figure 4 sensors-22-03587-f004:**
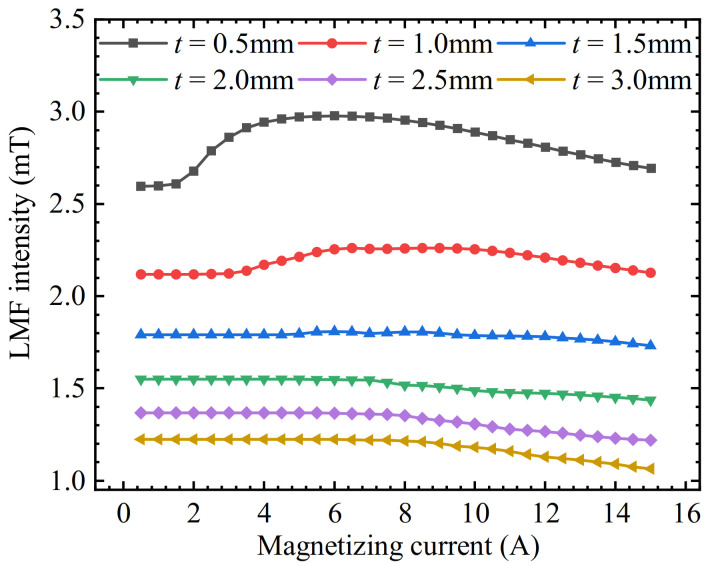
The relationship between the intensity of the defect LMF and the magnetizing current under different plate thicknesses *t*.

**Figure 5 sensors-22-03587-f005:**
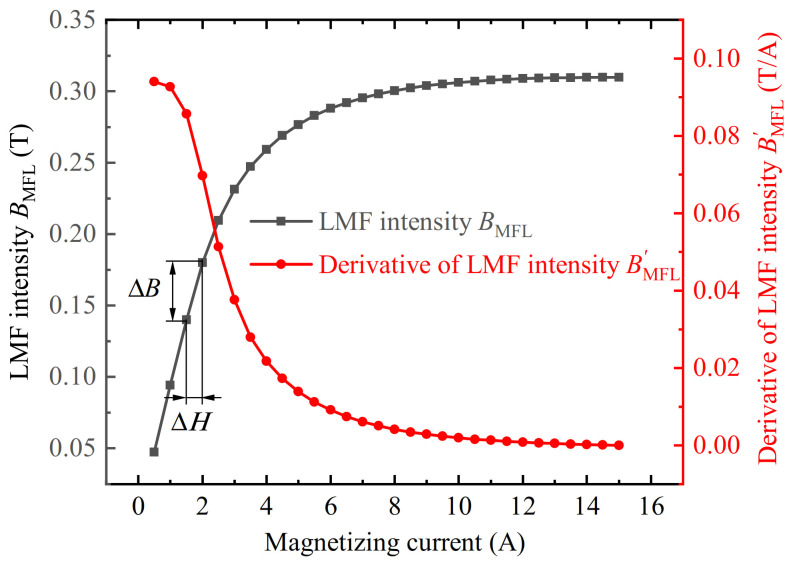
The change of the slot LMF intensity *B*_MFL_ and its derivative $B_{\mathrm{MFL}}^{\prime }$ with different magnetizing currents.

**Figure 6 sensors-22-03587-f006:**
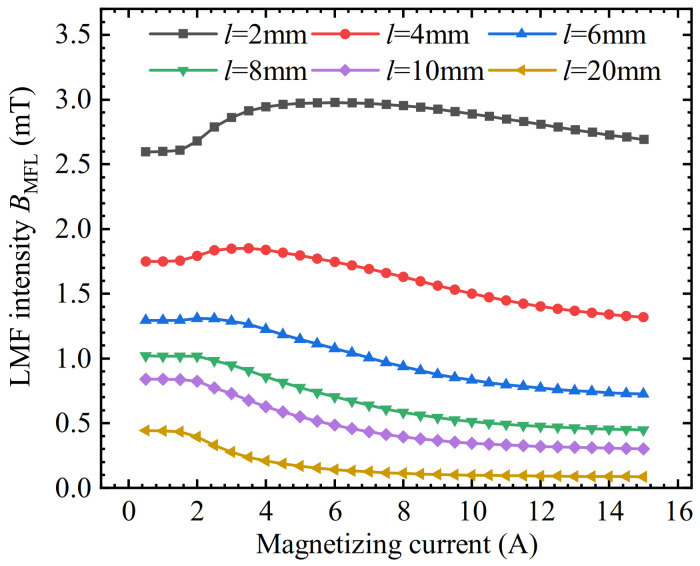
The relationship between the defect LMF intensity and the magnetizing current with coils of different lengths.

**Figure 7 sensors-22-03587-f007:**
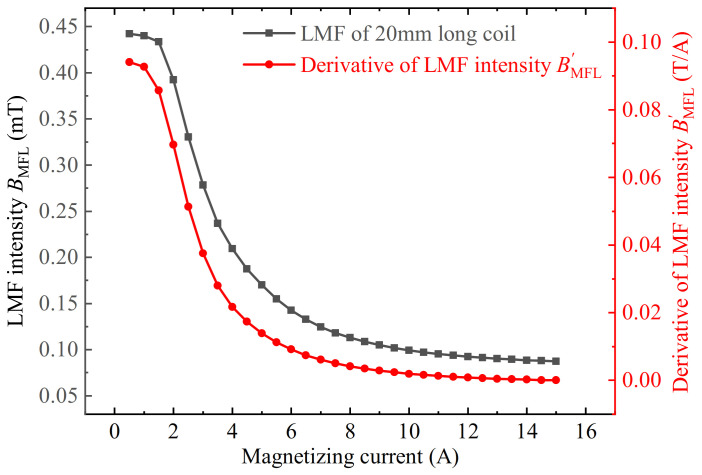
The comparison between the defect LMF intensity BMFL change trend of 20 mm long coil and the derivative of LMF intensity $B_{\mathrm{MFL}}^{\prime }$ with the magnetizing current.

**Figure 8 sensors-22-03587-f008:**
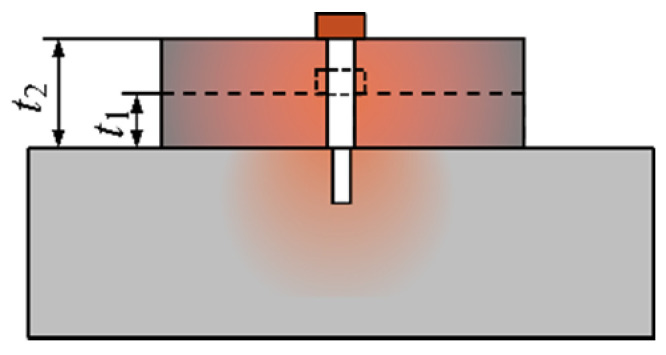
Schematic diagram of the thickness variation of SFLL.

**Figure 9 sensors-22-03587-f009:**
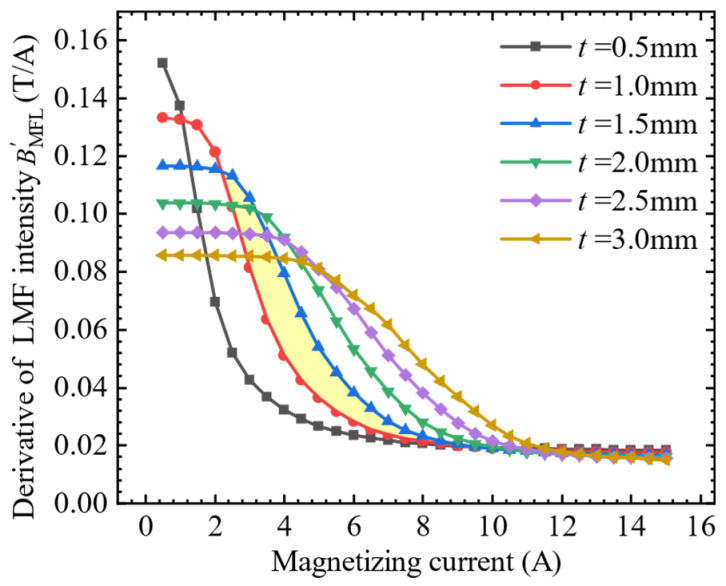
The relationship between the derivative of LMF intensity $B_{\mathrm{MFL}}^{\prime }$ and the magnetizing current with iron plates of different thicknesses.

**Figure 10 sensors-22-03587-f010:**
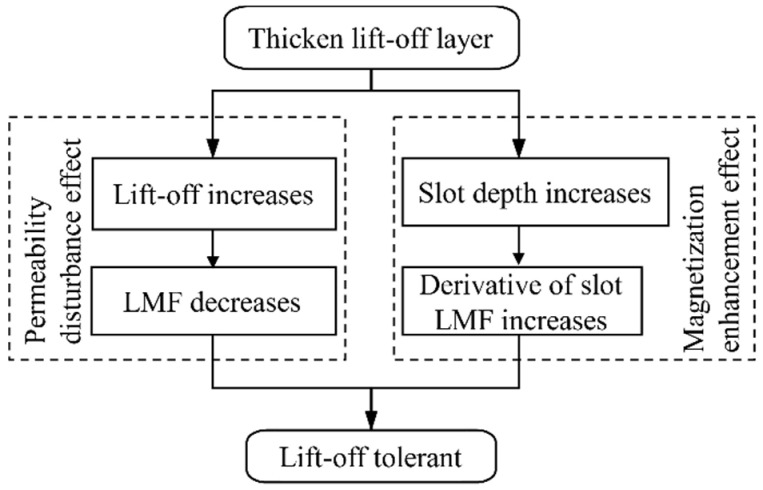
Schematic diagram of the lift-off tolerance of the SFLL-MFL method.

**Figure 11 sensors-22-03587-f011:**
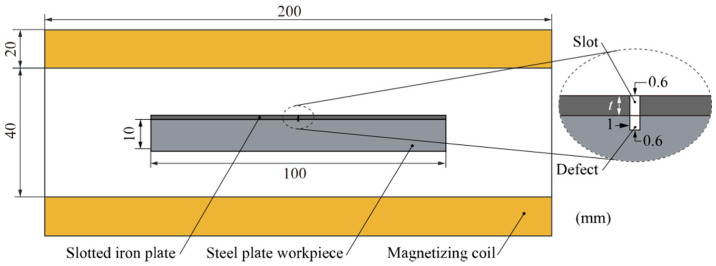
Two-dimensional simulation model of the SFLL-MFL method.

**Figure 12 sensors-22-03587-f012:**
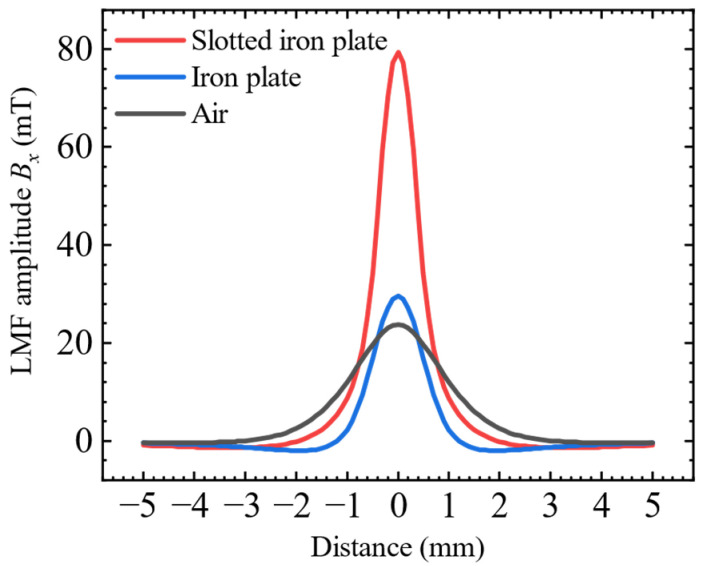
Comparison of LMF distribution between slotted iron plate, iron plate, and air lift-off layer.

**Figure 13 sensors-22-03587-f013:**
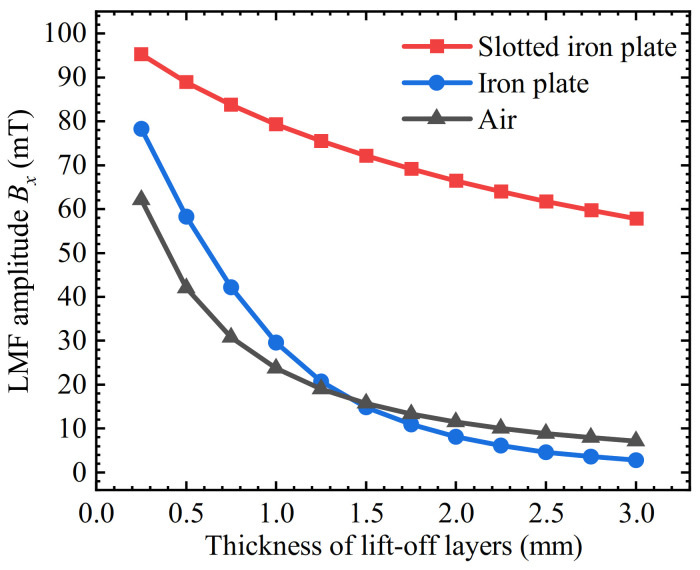
The relationship between the *B_x_* and the thickness of the lift-off layer for the slotted iron plate, iron plate, and air lift-off layer.

**Figure 14 sensors-22-03587-f014:**
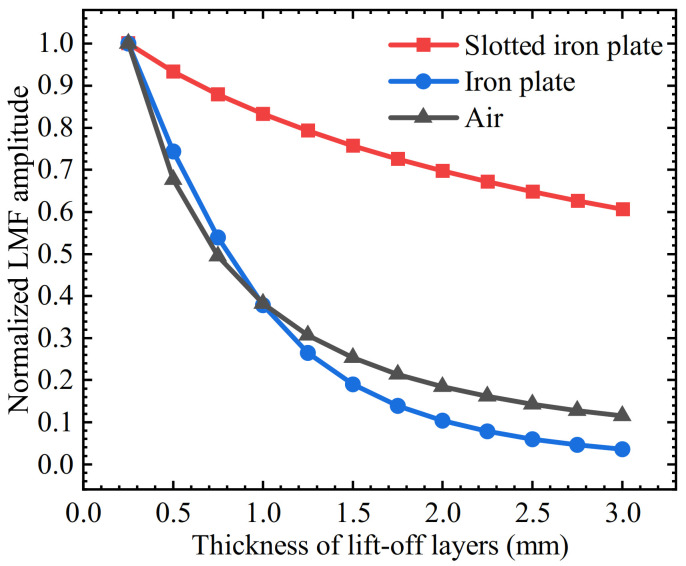
Relationship between the normalized LMF *B_x_* and the thickness of the lift-off layer for the slotted iron plate, iron plate, and air lift-off layer.

**Figure 15 sensors-22-03587-f015:**
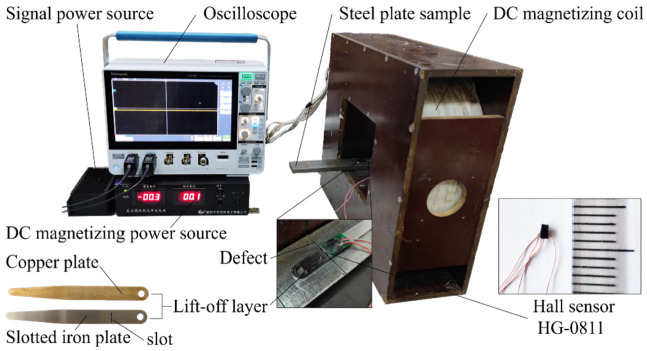
Schematic of the experimental setup for SFLL-MFL detection.

**Figure 16 sensors-22-03587-f016:**
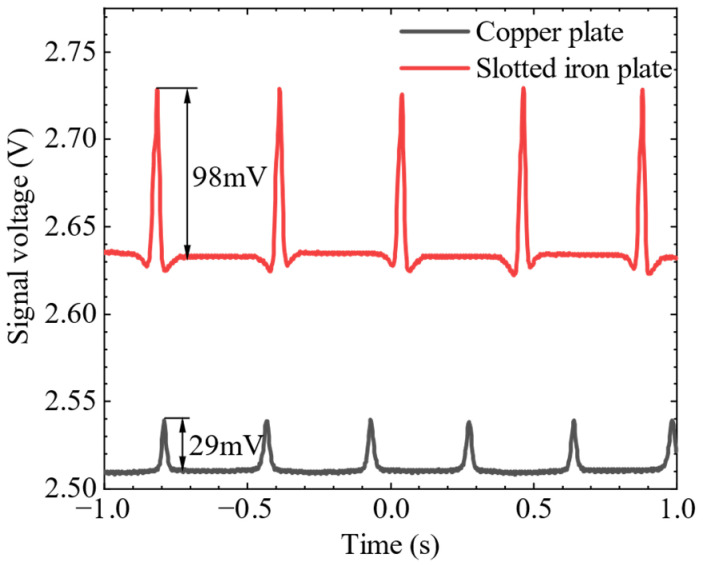
MFL signal in lift-off layer of copper plate and slotted iron plate.

**Figure 17 sensors-22-03587-f017:**
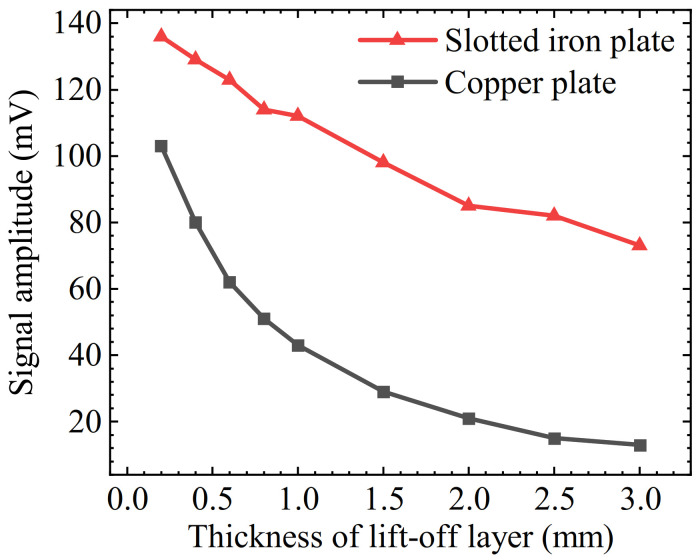
Relationship between the signal amplitude value of two kinds of lift-off layers and their thickness.

**Table 1 sensors-22-03587-t001:** The *B*-*H* curve of carbon steel Q235.

*H* (A/m)	*B* (T)
0	0
500	0.88529
1000	1.27739
1500	1.41839
2000	1.48692
2500	1.53552
3000	1.57709
4000	1.64878
5000	1.70842
6000	1.7579
7000	1.79885
8000	1.83341
9000	1.86285
10,000	1.88837
15,000	1.98345
20,000	2.05073
25,000	2.10193

**Table 2 sensors-22-03587-t002:** The parameters of the 2D simulation model.

Item	Parameters
Magnetization coil	Number of turns *N* = 3000, magnetization current *I* = 12 A
Steel plate workpiece	Material is Q235; length = 100 mm; plate thickness = 10 mm
Groove defect	Width = 0.6 mm; depth = 1.0 mm
Slotted iron plate	Material is Q235; thickness = 1 mm; slot width = 0.6 mm

**Table 3 sensors-22-03587-t003:** Comparison of attenuation speed with slotted iron plate, iron plate, and air lift-off layer under different thickness variation ranges.

Lift-Off Layer Thickness Range	Slotted Iron Plate	Iron Plate	Air
0.25 mm–1 mm	16.9%	62.1%	61.8%
0.25 mm–2 mm	30.2%	89.6%	81.6%
0.25 mm–3 mm	39.3%	96.3%	88.5%

**Table 4 sensors-22-03587-t004:** Comparison of amplitudes of the MFL signals and attenuation speed of slotted iron plate and air lift-off layer under different magnetization current *I* when the thickness of lift-off layer varies from 0.25 mm to 3 mm.

Magnetization Current *I* (A)	Air			SFLL		
	0.2 mm (mT)	3 mm (mT)	Attenuation Speed	0.2 mm (mT)	3 mm (mT)	Attenuation Speed
6	63.2	11.58	81.68%	90.22	52.38	41.94%
8	98.63	15.93	83.85%	122.8	66.79	45.61%
10	132.2	19.31	85.39%	150.5	75.96	49.53%
12	155.5	22.2	85.72%	174.6	84.73	51.47%
14	168.6	25.29	85.00%	193.3	91.62	52.60%

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
