# Peer review of "Mechanism of Magnetic Flux Leakage Detection Method Based on the Slotted Ferromagnetic Lift-Off Layer"

_sensors, 2022, doi:10.3390/s22093587_

Round 1
Reviewer 1 Report
The article deals with the use of the MFL method for non-destructive investigation based on the slotted ferromagnetic lift-off layer. The method provides a new idea for optimizing the design of the MFL sensor and improving the sensitivity of the high lift-off value magnetic flux leakage detection. the article has a logical structure, theoretical assumptions are based on results. Please incorporate my recommendations and comments.
To improve the detection sensitivity at large lift-off, many highly sensitive magnetic sensitive elements have been applied to MFL, such as high-sensitivity Hall element, anisotropic magnetoresistance (AMR), giant Magnetoresistance (GMR), and tunnel magnetoresistance (TMR). The letters that make up the shortcut should be in uppercase
Figure 3 .: please adjust the designation of quantities in the figure using the Italic font. / diameter = 23mm /
Figure 4. and other pictures - it is necessary to insert into the graph the functional values of the points from which the graph is constructed. Similar to Figure 5.
Figure 5: vertical axis labels / derivative of .... / would be more appropriate in the form of a mathematical expression.
Section 4: the title of the chapter needs to be changed: for example experimental set-up. ....
electron discharge machining (EDM): correctly it should be electric .....
Figure 15. The platform of experiments. I recommend editing the title of this image and at the same time adding a detail of the Hall sensor to the free space on the right side.
Conclusions: in conclusion, it is necessary to supplement the most significant results achieved, in the form of specific data.
The DC magnetizing coil with 3000 turns was powered by a DC magnetizing power: please add word power SOURCE.
Reviewer 2 Report
This manuscript reports a magnetic flux leakage detection method using a ferromagnetic lift-off layer with a rectangular slot. This method considers a magnetic sensor located above the slot to scan the workpiece together with the lift-off layer. Based on simulation and experimental results, the proposed method enhances the amplitude of the magnetic flux leakage signal in comparison with the conventional method that uses non-ferromagnetic lift-off layer. Thus, the proposed method could improve the sensitivity of the large lift-off value magnetic flux leakage detection.
1.- Introduction should add more information of the limitations of the conventional methods for magnetic flux leakage detection.
2.-Introduction should include the main advantages and results of the proposed method in comparison with other methods reported in the literature.
3.-Authors should incorporate more technical information about the simulation model such as magnetic properties of the materials, dimensions of the different components, mesh type, boundary and load conditions, and analysis type.
4.-Authors could include more discussion of the main simulation and experimental results.
5.-Authors could add more experimental results to study the response of the proposed method. This manuscript could incorporate discussions that compare the simulation results with respect experimental results.
6.-Which are the main limitations of the proposed method?
7.-What is the future research work?
Round 2
Reviewer 2 Report
Authors have improved their manuscript considering the comments of reviewer.